# Efficacy and Safety of COVID-19 Vaccines—An Update

**DOI:** 10.3390/diseases10040112

**Published:** 2022-11-23

**Authors:** Eshani Sharma, Sraddha Revinipati, Saisha Bhandari, Sejal Thakur, Shubham Goyal, Aruni Ghose, Sukrit Bajpai, Waleed Muhammad, Stergios Boussios

**Affiliations:** 1Department of Internal Medicine, Kasturba Medical College, Mangalore 575001, India; 2MGM Institute of Health Sciences, Navi Mumbai 410209, India; 3Department of Internal Medicine, Post Graduate Institute of Medical Education and Research, Chandigarh 160012, India; 4Department of Infectious Diseases, Kasturba Medical College, Manipal 576104, India; 5Department of Internal Medicine, Newham University Hospital, Barts Health NHS Trust, London E13 8SL, UK; 6Department of Medical Oncology, Medway NHS Foundation Trust, Windmill Road, Gillingham ME7 5NY, UK; 7Department of Respiratory Medicine, Queen Elizabeth Hospital, University Hospitals Birmingham NHS Foundation Trust, Birmingham B15 2TH, UK; 8Faculty of Life Sciences & Medicine, School of Cancer & Pharmaceutical Sciences, King’s College London, London SE1 9RT, UK; 9AELIA Organization, 9th KM Thessaloniki—Thermi, 57001 Thessaloniki, Greece

**Keywords:** COVID-19, SARS-CoV-2, vaccine, safety, efficacy

## Abstract

A few centuries ago, the first vaccine vial was formulated, and since then, they have resulted in an eminent reduction in infectious diseases associated morbidity and mortality. The discovery of the novel SARS-CoV-2 virus and the COVID-19 disease and its steady progression to a global pandemic with 603,711,760 confirmed cases and 6,484,136 reported deaths according to the World Health Organization (WHO) on 7 September 2022 was exceedingly catastrophic. This brought about an unexpected need for preventative and cost-effective measures to curb the devastating impact of the virus, followed by accelerated competition within the pharma giants to manufacture and dispense vaccines at an exponential rate. Non-pharmaceutical medications such as mandated face mask policies, the imposition of travel limitations and generalized disinfectant use were somewhat successful in mitigating the catastrophic effect, but the onus fell upon vaccination strategies and other medical interventions to counteract and subdue this international health threat. The need to ensure current and future pandemic preparedness, however, presents multiple hurdles, among which are equitable vaccine access and the rising trend of vaccine hesitancy at an individual and international level, which are beyond the scope of this discussion. With this review article, we seek to draw perspective on current COVID-19 virus variants, in-hand vaccine types with their mechanism of action along with their effectiveness and safety profile. We also aim to discuss substantial side effects while adding a segment on the booster dose controversy.

## 1. Introduction

Towards the end of December 2019, the Coronavirus Disease (COVID-19), caused by the Severe Acute Respiratory Syndrome Coronavirus 2 (SARS-CoV-2), originated in Wuhan, Hubei province, China [1]. A pandemic was declared by the World Health Organization (WHO) on 11 March 2020, and as of 7 September 2022, a total number of 603,711,760 cases and 6,484,136 casualties were reported worldwide [2]. COVID-19 affected healthcare systems worldwide at every level [2,3]. The disease spectrum of this pathogen ranges from a mild self-limiting infection to potentially fatal disease with inflated morbidity and mortality figures for individuals with coexisting morbidities such as diabetes mellitus, obesity and other lifestyle diseases, along with poorer outcomes in patients with respiratory comorbidities, such as COPD [4,5].

Akin to other RNA viruses, SARS-CoV-2 is also susceptible to genetic variation and mutation, leading to the development of multiple variants. The WHO formulated the Technical Advisory Group on Virus Evolution (previously referred to as WHO Virus Evolution Working Group) to name and characterize the evolving variants of the virus, which have been listed below [6].

## 2. Variants of Concern (VOC)

### 2.1. Current VOC

B.1.1.529 lineage—Alias for the Omicron variant; it was initially reported in South Africa at the Lancet Laboratory during the November of 2021. Its molecular structure comprises numerous mutations especially of the spike protein (S) which plays a vital part in improving the infectivity, transmissibility as well as the immune evasion for the individual [7,8].

### 2.2. Past VOC

B.1.1.7 lineage—Alpha variant or GRY was discovered initially in the United Kingdom in December of 2020. It was shortlisted as a VOC due to the S-gene target failure in PCR samples, as well as the presence of 17 mutations in its genome. Significance of these changes is observed by the markedly raised affinity of the spike protein to ACE-2 receptors augmenting attachment and raising COVID-19 severity as compared to the other varied variants in circulation [9].

B.1.351 lineage—The Beta variant was initially discovered in South Africa in October of 2020 [10]. The viral genome contains nine mutations in spike protein alone. It has an increased transmission risk and reduced success in management with convalescent sera and monoclonal antibody therapy [11].

B.1.1.28.1 lineage—In December 2020, Brazil reported the Gamma variant [12]. It bears ten mutations on spike protein, three of which were rather homogenous to VOC B.1.351. This variant also decreases the reaction to monoclonal antibody therapy and the convalescent sera [11].

B.1.617 lineage—The Delta variant was initially discovered in India in October of 2020. This VOC shows a genetic variation at the receptor binding dominion of the spike protein, which raised the affinity of the virus to bind to ACE2 receptors, providing it with far more advanced transmissibility [12,13].

## 3. Variants of Interest (VOI)

### 3.1. Current VOI

At present, there is no VOI in circulation.

### 3.2. Past VOI

This category includes variants such as Epsilon, Eta, Iota, Kappa, Lambda and many others, which were classified as such because of their potential to show a reduced response to an antibody or vaccine sera management [6].

In a background of aggressive disease and treatment failure, prevention might be the strategy that could successfully redeem stability in global health, the key to which is the rapid development, manufacturing and distribution of vaccines. Since the formulation of the first vaccine vial a few centuries ago, vaccines have progressively led to a remarkable decrease in the morbidity and mortality caused by various infectious diseases [14]. In this vein, as of 12 September 2022, a total of 12,613,484,608 vaccine doses have been administered [2].

In this article, we examine the efficacy and safety of various vaccines against COVID-19 made available by a number of organizations worldwide. According to the WHO, in September 2022, 199 vaccine candidates were in the preclinical phase, and 172 vaccines were registered in clinical trials. Among the clinical applicants, 54 vaccines are in phase I (Testing safety and dosing), 15 vaccines in phase II (expanded safety trials), 45 vaccines in phase III (large-scale efficacy testing) and 11 vaccines have been granted Emergency Use Listing (EUL) [15].

## 4. COVID Vaccines: Types and Mechanism of Action

As discussed above, the various strains emerge due to mutations mostly in the ACE-2 receptor binding site, the receptor binding domain (RBD) and the N-terminal domains. This encouraged the formulation of many vaccines with distinct working principles, mainly divided into four groups.

### 4.1. mRNA Vaccines

These vaccines use synthetically manufactured mRNA, which infects the host cells and produces a component of S protein. The body then degrades it while the protein triggers the production of antibodies. These antibodies prepare the body to tackle any future infection with minimal risk of adverse effects. The vaccines using this mechanism are Pfizer and Moderna, produced by BioNTech and Sanofi, respectively [16]. BNT162b2 developed by Pfizer/BioNtech elicits an immunological response by inducing IgG, IgA, CD8+ cells, or CD4+ cells, while mRNA-1273 developed by Moderna induces CD8 T cell response [17].

### 4.2. Viral Vaccines

These are modified versions of a virus belonging to a different genus used as a vector. It interacts with the immune cells and aids them in acknowledging and outwitting the pathogenic virus. Once injected, the immune cells of the body detect the presence of foreign antigen and activate an immune response by producing antibody-producing B cells and T cells that seek out and destroy infected cells. T cells act by examining the storage of proteins expressed on the surfaces of cells. Since they can recognize the body’s own proteins as ‘self’, if they find a foreign protein, they activate an immune response against the cell storing it [18]. However, their use is limited due to the increased risk of side effects. As of April 2021, the vaccines available using this mechanism are the Janssen, AstraZeneca and Sputnik V vaccines produced by Johnson & Johnson, Oxford-AstraZeneca PLC and Panacea Biotec, respectively [19].

### 4.3. Whole Pathogen Vaccines

One of the most prevalent and age-old established vaccines is currently of two types—live attenuated and inactivated. The inactivated vaccines are prepared by destroying the virus’s genetic material with chemicals, heat and radiation. Since these vaccines are versions of weak natural pathogens, the immune system activates a range of defenses such as killer T cells which identify and destroy infected cells, helper T cells which support antibody production and antibody-producing B cells which will target pathogens [20]. When introduced into the body, they stimulate antibody-mediated responses, which are weak and relatively less long-lived. Thus, they are always administered along with an adjuvant and booster doses are often required. In contrast, the live attenuated vaccines use a weakened form of the virus in the body. When introduced into the body, these viruses can grow and replicate within the body but cannot cause symptomatic disease in the individual [21]. The vaccines using the inactivated mechanism include Covaxin, Sinopharm, Corovac and Sinovac vaccines. The live attenuated counterparts are yet in the trials phase; nevertheless, one such vaccine approved is Covivac [22].

### 4.4. Protein Subunit Vaccines

These vaccines, unlike whole virus vaccines, use specific parts of the virus-like fragments, antigens, parts of protein or polysaccharides which are incapable of producing any sort of infection to the body [23]. These fragments have pathogen-associated molecular patterns, which are recognized by the pattern recognition receptors, principally the Toll-like receptors. Upon engagement of the above, the intracellular signaling cascade is triggered. This leads to the release of various proinflammatory molecules, which orchestrates the building of adaptive immunity [24]. The absence of such pathogen-associated molecular patterns would lead to a weaker immune response. Hence, they are given adjuvants as the antigens are insufficient to induce long-term immunity. The vaccines using this mechanism are the Sanofi-GSK, Novova and Dynavax.

Figure 1 depicts a timeline of the major COVID vaccines, whereas Figure 2 depicts the mechanism of action of various types of vaccines.

## 5. Efficacy of COVID Vaccines

NPIs (non-pharmaceutical interventions) such as mandatory face masks, national and international travel limitations and marked disinfectant use were somewhat successful in subduing global healthcare downfall, the onus fell upon vaccination strategies to counteract this international threat [26]. While the levels of neutralizing antibodies do not provide a direct measure of vaccine efficacy against the varied variants of SARS-CoV-2 and certainly do not paint the entire picture, barring the effects of T-cell immunity, complement system, it gives an elementary idea of their efficacy in the real world.

The following section highlights the effectiveness against specific vaccines against different variants of the virus. It is important to mention at this juncture, that the information should be interpreted with caution as due to the urgency of developing vaccines for this morbid disease, several studies even today are lacking adequate sample size, randomization or distribution over ages and remain quite preliminary.

### 5.1. NVX-CoV2373

B.1.1.7 variant—Efficacy trials undertaken by the biotech firm, Novavax revealed that their vaccine was highly effective in producing antibodies for the B.1.1.7 variant of COVID-19 discovered in the UK [27]. Shen et al. compared the B.1.1.7 variant to the D614G in neutralization assays with the serum samples collected from 28 people who received NVX-CoV2373 two weeks after the second dose [28].

B.1.351 variant—Shinde et al. conducted a randomized, double-blind controlled trial among 4387 recipients, which relayed that when participants had been given two doses of the NVX-CoV2373 vaccine, it showed an efficacy of 49.4% (95% (confidence interval) CI, 6.1–72.8) against a COVID-19 infection caused by the variant B.1.351 [29].

B.1.1.529 variant—There is scarce data on the particular effectiveness of NVX-CoV2373 on Omicron. A preprint study reported a decreased response to Omicron compared to Delta and other variants on primary vaccination but cross-reactive antibodies on a three-dose booster regimen [30].

### 5.2. Ad26.COV2.S

D164G mutation and original Wuhan-Hu-1—Sadoff et al.’s conduction of a randomized, double blind, placebo controlled phase-three trial relayed that Ad26.COV2.S showed immunity against moderate to severe COVID-19, 14 days post dose. In a study population of 19,630 receiving the vaccine, an efficacy of 66.9% (95% CI, 59.0–73.4), which remained 28 days after the dose at around 66.1% (95% CI, 55.0–74.8), was reported. Effectiveness against severe COVID-19 has risen to 76.7% (95% CI, 54.6–89.1) for ≥14 days and 85.4% (95% CI, 54.2–96.9) for ≥28 days [31].

B.1.351 variant—In a trial with 19,630 participants who received the vaccine in South Africa, 94.5% of the sequences were of the B.1.351, and vaccine efficacy sustained a 52.0% in moderate, severe as well as critical conditions of COVID-19 and 73.1% in severe to critical COVID-19 disease ≥14 days after administration. At ≥28 days after administration, efficacy had risen to 64.0% in moderate to severe COVID-19 disease and 81.7% in severe COVID-19 disease. In samples collected in Brazil, 69% showed P.2 lineage carrying the E484K mutation. Despite infection from varied variants, the COVID-19 vaccinations, formulated on the Wuhan-Hu-1 strain, vaccine efficacy remained high. From this, it can be inferred that these vaccines show a cross-protective efficacy with the new variants in SA and Brazil [31].

B.1.1.529—During the Omicron surge, a study was conducted on the population in South Africa in which 162,637 PCR tests were analyzed, of which 93,854 (57.7%) were taken from recipients of both the doses of the BNT162b2 vaccine which were administered forty-two days apart from each other or two doses of the Ad26.COV2.S vaccine which were administered four to six months apart from each other. Within this fraction, of the 34% that were positive, 1.6% underwent admission to a hospital and 0.5% were critical with ICU admission. Of the recipients of the Ad26.COV2.S vaccine, immunity against hospitalization for the disease showed 55% (95% CI, 22–74) after the second dose in less than 13 days, 74% (95% CI, 57–84) at 14 to 27 days, and 72% (95% CI, 59–81) at 1 to 2 months; protection against ICU admission was 69% (95% CI, 26–87) at 14 to 27 days and 82% (95% CI, 57–93) at one to two months post the administration of the second dose [32].

### 5.3. BNT162b2

D164G mutation, original Wuhan-Hu-1, B.1.1.7, B.1.351—Tada et al. took sera sampling from people who underwent vaccination with BNT162b2 and analyzed their neutralizing activity against D164G mutation strain, B.1.1.7 lineage and B.1.351 lineage spike proteins. Serum samples of vaccinated individuals neutralized D164G strain with seven-fold raised antibody titer than convalescent serum [33]. Sera which neutralized the virus with B.1.1.7 spike protein showed equivalent results, which relayed that the vaccine provides a raised immunity against this variant. There was a threefold reduction in the titer when the sera neutralized the virus with B.1.351 spike protein. This can be credited to the E484K mutation present. Correspondingly, a study using a B.1.1.7 pseudo virus showed the vaccine to remain effective against variants with a slight decrease in neutralization [34]. A study conducted in Qatar, a case–control study of 265,410 persons having received the two-dose regimen, relayed that the vaccine’s efficacy against the B.1.1.7 variant was 89.5% (95% CI, 85.9–92.3) <14 days post the 2nd dose. The efficacy against the B.1.351 variant was 75.0% (95% CI, 70.5–78.9) [35].

P.1 lineage—Dejnirattisai et al. studied the antibody evasion of the P.1 strain. Sera from 25 BNT162b2 vaccinated individuals were used. They concluded that the neutralization titers against P.1 strain were similar to that of B.1.1.7, thus inferring adequate cross-protection of vaccinated individuals against it. Another observation was that B.1.351, or the South African Variant, had a maximum reduction in titer [36].

B.1.617.1 and B.1.617.2 lineage—Liu C et al. compared the reduction in neutralization of variants B.1.617.1&2. There was a reduction of 2.7-fold in the sera of 20 BNT162b2 vaccinated individuals for B.1.617.1 and a 2.5-fold decrease for B.1.617.2. These reductions were similar to those of B.1.1.7 and P.1, indicating that there is no sizable escape in neutralization, unlike the reduction seen in B.1.351 [37]. In a similar neutralization assay study, Liu J et al. reported a modest neutralization reduction when compared to wild-type virus of B.1.617 lineage, the reduction more powerful in B.1.617.1 strain. Although reduction was noted, BNT162b2 immune sera still competently neutralized all strains of this lineage [38]. Zani A et al. studied the neutralization of B.1.525 lineage in addition to other variants. They reported that the virus of said lineage was sufficiently neutralized by the sera of 37 BNT162b2 vaccinated people, as compared neutralization of B.1.1.7 and D164G mutation lineage [39].

### 5.4. BBV152A

B.1.1.7—The neutralization assay study by Sapkal et al. using sera of individuals vaccinated with BBV152A showed that the escape of the UK variant from this vaccine is unlikely [40].

B.1.351 and B.1.617.2—Yadav et al., in their study of neutralization of B.1.351 and B.1.617.2 variants with the serum of 20 recipients of BBV152 A, reported reduction in titers with these variants, but demonstrated that the neutralizing potential of the vaccine remains well established [41].

B.1.1.28.2—The results of yet another study conducted by Sapkal et al. revealed a 1.92-fold reduction in neutralization titers, when compared to D164G mutation lineage [42].

B.1.1.529—In a study that explored vaccine efficacy against this VOC, virus shedding and lung viral load, along with less morbid disease were relayed in the vaccinated cohorts as compared to the placebo groups. Presently, it is found that a COVAXIN^®^ booster dose augments efficacy of the vaccine against the Delta COVID-19 disease and also offers immunity against morbidity caused by the Omicron variant [43].

### 5.5. mRNA-1273

B.1.1.7—In a study in Qatar, with 181,304 recipients of the full two-dose regimen, the calculated effectiveness against infection with B.1.1.7 was scarce for the initial two weeks after the first dose, which rose markedly to 81.6% (95% CI, 73.1–87.8%) in the 3rd and 94.4% (95% CI, 89.1–97.5%) in the 4th week and attained 99.2% (95% CI, 95.3–100.0%) in the 2nd week after the subsequent 2nd dose, whereas it was 100% (95% CI, 91.8–100.0%) post 14 days of the second dose [44].

B.1.351—In the above-mentioned study, PCR positive samples of the B.1.351 variant was also collected, and the effectiveness against COVID 19 with B.1.351 was scarce for the initial 2 weeks after the initial dose but markedly raised at the 3rd week to acquire a 47.9% (95% CI, 39.5–55.2%). Efficacy was 73.7% (95% CI, 67.6–78.8%) in the 4th week prior to the 2nd dose and attained 96.4% (95% CI, 94.3–97.9%) in the 2nd week post the 2nd dose and 96.4% (95% CI, 91.9–98.7%) post 14 days of the second dose [44].

B.1.1.529—As one would expect, any strain showing minor changes in spike proteins showed increased escape from humoral response induced by primary (two dose) vaccination. Omicron proved to be no different, due to its highly mutated spike proteins, it escaped neutralization in a large proportion of people who had received two doses of mRNA-1273. Interestingly though, people who had received a booster in the three months prior to a study conducted by Garcia-Beltran WF et al., there were cross neutralization responses to Omicron, thus indicating that a booster of mRNA-1273 increased protection against this VOC mRNA-1273 and BNT162b2 vaccines offer a significantly better response than the previously established J&J vaccines [45,46].

### 5.6. AZD-1222

B.1.1.7—The vaccine showed reduced titers for neutralization with this lineage, but clinical efficacy against features of the disease caused by variant B.1.1.7 was adequate, suggesting that low neutralizing antibody titers are adequate to provide protection [47].

B.1.351—A randomized control trial, with 2026 participants, conducted in Africa concluded that 2 doses of this vaccine had no effect against mild to moderate disease caused by B.1.351, but notably there were no reports of morbidity from severe disease either. This reduced efficacy should be considered, with the background that the first dose of this vaccine had 75% efficacy (95% CI, 8.7 to 95.5) in protecting against mild to moderate infection of COVID-19 before the emergence of this variant of concern [48].

B.1.617—The geometric mean neutralization titers against B.1.617.1 were 2.7 times less when compared to the Victoria virus for the Pfizer-BioNTech vaccine serum and 2.6 times less for the Oxford-AstraZeneca vaccine. The reductions were commensurable in scale with those seen with B.1.1.7 and P.1, with no indication of broad abdication from neutralization, contrary to what is seen with B.1.351. From this, we can conclude that the current vaccine is sufficient to provide protection against severe infections caused by the B.1.617 variant, although parallelly an increase in breakthrough infections can be expected [37].

B.1.1.529—In a retrospective analysis, 886,774 Omicron-infected individuals were identified and effectiveness studied; it was found that after two doses of the AZD-1222 vaccine, no effect was seen. The efficacy after an AZD-1222 primary course had increased to 70.1% (95% CI, 69.5–70.7) after 2–4 weeks of an mRNA-1273 booster and reduced to 60.9% (95% CI, 59.7–62.1) after 5–9 weeks [45].

### 5.7. Sputnik V

The provisional outcomes of the past 3 Gram-COVID-Vac trials, in which 19,866 received two doses of the vaccine, exhibit that the vaccine is 96.1% (95% CI, 85.6–95.2) effective against disease caused by SARS-CoV2. This includes the period 21 days after the 1st dose to the day of receiving the 2nd dose. However, unlike some other vaccine candidates, the outcomes of the vaccine were not 100% (95% CI, 94.4–100) effective against the severe form of COVID-19; The results were preliminary as this was a secondary outcome [49].

### 5.8. CoronVac

In a double-blind, placebo-controlled, randomized phase I/II clinical trial by Zhang Y et al., examining the safety and efficacy of CoronaVac on healthy adults aged 18–59 (743 participants received at least one dose), it was reported that CoronaVac was well tolerated and induced adequate immune response against a COVID-19 infection. According to the trial, protective efficacy is yet to be determined. The studies on the efficacy of the vaccine against different strains are lacking [50].

### 5.9. Sinopharm

According to Huang et al., the 501Y.V2 variant remains under the umbrella of protection offered by vaccines targeting the whole virus (BBIBP-CorV). The potential 1.5–1.6 times reduction in neutralizing GMTs must be considered for their effect on the clinical efficacy of these vaccines. For these vaccines, immune serum samples neutralize both variants 501Y.V2 and D614G [51].

Two case studies from Brazil reported breakthrough infection in two vaccines: in one, 122 days following administration of the second dose, and in the other, 106 days post administration of the second dose. Both were reported to be infected by the P.1 variant of SARS-CoV2. Both patients recovered fully and did not develop sequelae or severe illness [52].

## 6. Safety and Adverse Effects of Vaccines

All the available vaccines for COVID-19 are phase III approved and hence are considered safe. However, certain side effects were observed in individuals after both of the doses and hence some contraindications were provided.

### 6.1. Novavax

A study conducted by Health T et al. explored the safety and efficacy of NVX-Co2373, involving about 7500 individuals receiving the vaccine found various adverse effects after both first and second dose, ranging from mild, but more common symptoms such as fatigue, headache, muscle pain, to more serious systemic adverse events such as a fever (between 38 °C and 40 °C in 2.4% participants after the first dose, and 5.4% after the second dose) of >40 °C in two participants, one after the first dose and one after the second. Of note, one serious adverse event, myocarditis, was reported in a participant 3 days after the second dose, identified as a possible immune-mediated adverse event. Adverse events were less serious and occurred less frequently in older participants in the trial. Due to the number of participants, it is not possible to exclude the occurrence of the rare adverse events [53].

### 6.2. Pfizer and Moderna

An article comparing the pharmacology, indications, contraindications and adverse effects of both the Pfizer/BioNTech and Moderna vaccines found that both of these vaccines were advantageous in providing immunity in the case of COVID-19. They are recommended for people over the age of sixteen and provide immunity for the next 119 days after the initial vaccination; they are 95% effective in the prevention of COVID-19. The Moderna vaccine is recommended for people over the age of eighteen. It provides immunity for the next 119 days after the initial vaccination and is 94.t% effective in the prevention of COVID-19. Adverse effects seen were allergic symptoms associated with both the vaccines such as redness, swelling at the site of vaccine, muscle and joint pain, fever, fatigue, headache, nausea, vomiting, itching, and chills, rarely even causing an anaphylactic shock. The occurrence of adverse effects was lower in the Pfizer/BioNTech vaccine when compared to Moderna vaccine; however, the storage and transport of the Moderna vaccine is easier because it is less temperature sensitive [54]. Of note, a rare, but serious adverse event was noted in the population of the United States, within its vaccine adverse events reporting system, catching 1226 reports of myocarditis after either dose of these mRNA vaccines, between December 2020 and June 2021, with acute hospitalization, but no casualties. This led to a review by the CDC, but the benefits of the mRNA vaccines remained greater than the risks, and their use continued [55].

### 6.3. CoronaVac

Another study investigated CoronaVac—an inactivated vaccine candidate against COVID-19—for its safety, tolerability and immunogenicity in China. It was noted that the two-dose regimen of the vaccine at different concentrations at different dosing schedules did not cause major side effects in healthy adults aged 18–59 years, the most common adverse effect being pain at the site of injection, similar to previous findings for another inactivated COVID-19 vaccine from Sinopharm (Beijing, China). Hence, compared to COVID-19 vaccine candidates with other mechanisms of action, such as viral-vectored vaccines or DNA or RNA vaccines, it was found that the occurrence of fever after vaccination with CoronaVac (inactivated vaccine) was relatively low, less than 20% [50].

### 6.4. Johnson & Johnson (J&J)/Janssen

A rare side effect was seen in a small group of women treated with the J&J/Janssen’s vaccine. Thrombosis with thrombocytopenia syndrome was found at a rate of about 7 cases per 1,000,000 vaccinated women between 18 and 49 years old and rarer still in women over the age of 50. A review showed that the J&J vaccine’s known and potential benefits exceeded its known and potential drawbacks. Although this condition occurred rarely and is treatable when diagnosed in time, thrombosis with thrombocytopenia syndrome is a severe condition. Other symptoms associated with this vaccine are severe or persistent headaches or blurred vision, shortness of breath, chest pain, leg swelling, persistent abdominal pain, easy bruising or tiny blood spots under the skin near the injection site [56].

### 6.5. Ad5 Vectored COVID-19 Vaccine

A non-randomized phase I trial of an Ad5 vectored COVID-19 vaccine was conducted in Wuhan, China to evaluate the safety and tolerability of the individuals to the vaccine. On 16 March 2020, 195 healthy adults aged between 18 and 60 years were registered and assigned to one of three dose groups to receive an intramuscular injection of vaccine. Of the cohort of hundred and eight participants, some of them received a lower dose (n = 36), a middle dose (n = 36) and a higher dose (n = 36) of the vaccine. There was at least one adverse reaction relayed in less than a week in thirty (83%) of the cohort in the middle dose category, and twenty-seven (75%) in the high dose category. The most common injection site reaction was pain which was relayed in 58 (54%) of the cohort, while the most commonly related systemic reactions were fever (46%) followed by fatigue which was seen in 44%, headache in 39% and muscle pain in 17%. Most of the adverse reactions relayed were mild-to-moderate in severity. There were no serious reactions reported twenty-eight days post-vaccination [57].

### 6.6. AstraZeneca and COVISHIELD

It is noted that AstraZeneca (vector vaccine) and COVISHIELD COVID-19 vaccines cause a rare form of blood clot after vaccination called Vaccine-Induced Immune Thrombotic Thrombocytopenia (VITT), occurring 4–28 days after vaccination. Although VITT seems to be rare, it is estimated to occur in between 1 per 26,000 and 1 per 100,000 persons vaccinated with a first dose of AstraZeneca or COVISHIELD COVID-19 vaccine, globally. In Canada, it is estimated to be approximately 1 per 55,000 doses. This situation is being closely observed in Canada and internationally. Although very rare, cases of capillary leak syndrome—a condition that causes fluid leakage from the capillaries—has also been reported following vaccination with AstraZeneca or COVISHIELD; hence, individuals with a history of capillary leak syndrome should not receive these vaccines [58]. A more serious adverse event has been described in a case series by Schultz et al., where they reported a rare vaccine related event, vaccine induced thrombotic thrombocytopenia, where 5 health workers within a study population of 130,000 developed serious venous embolism and concurrent thrombocytopenia 7–10 days after the first dose of the vaccine [59].

### 6.7. Sputnik V

To understand the side effects and safety of this vaccine, an example can be drawn from the study conducted in Russia, which explored the pattern of incidence of symptoms and determined how doses, age, gender and sequence of shots could show adverse effects and safety doses. The data collected was a unique dataset consisting of 11,515 self-reported Sputnik V vaccine users who reported adverse effects on a Telegram group. Options of adverse effects included pain, lymph node enlargement, swelling, erythema, pruritus, headache, fever, chills, fatigue, nausea/vomiting, diarrhea and insomnia. The results of this retrospective analysis showed that the adverse effects reported were higher in females than in males (1.2-fold, *p* < 0.001) and more in the first dose than the second (1.13-fold, *p* < 0.001), and the number of effects decreased with age (*β* = 0.05 per year, *p* < 0.001). The results also concluded that Sputnik V adverse effects were more similar to other vector vaccines (132 units) administered when compared with mRNA vaccines administered (241 units) [60].

### 6.8. Sinopharm

In Iran, patients who have multiple sclerosis were among the first to receive the vaccination in Iran. The majority of the recipients received the Sinopharm COVID-19 vaccine. The study was conducted on 583 patients between 1 May 2021 and 22 May 2021. No serious adverse effects were reported. Most common complaints were constitutional symptoms, such as malaise, fatigue, fever, shivering and generalized body pain (51%) and headache (9%). Only five recipients (0.9%) reported multiple sclerosis relapse after vaccination. Hence, the conclusion was drawn as safe for use to these patients [61].

## 7. COVID Vaccines in Specific Populations

The conception of the COVID-19 pandemic was a monumental threat not only to the general population but determinately to a certain sub-section of the population, those with chronic diseases which include autoimmune conditions, those with ongoing immunosuppression or those undergoing cancer treatment. The patients who are immunocompromised were not recommended to take live attenuated vaccines as the possibility of infection was relatively higher in these patients. Similarly, in those patients undergoing cancer treatment, live vaccines were not advised, vaccination against COVID-19 for all cancer patients was recommended and deemed as safe and effective except for those who are currently on anti-B cell therapies; in such patients, an interval of four to six months is advised for vaccination post cessation of medication. It is recommended for all those patients who are scheduled for solid organ transplants to be vaccinated before the transplantation; post-transplantation, a minimum of a three-month interval should be followed before vaccination to avoid acute rejection.

While considering autoimmune conditions, specifically multiple sclerosis, vaccination is advocated primarily without cessation of treatment keeping in mind that there could be an imminent risk of worsening or relapse of the disease following discontinuation of therapies. In irritable bowel syndrome (IBS), vaccination is recommended for all patients, live attenuated vaccines are not proffered and for those patients with acute presentations, a gap is recommended before vaccination, when the patient is taking a lower dose of corticosteroids [62].

There has been a reported difference in the immune response to the BNT162b2 vaccine post the first dose. In comparison, the elderly (prominently in >80 years) showed a significantly lower level of antibodies than the younger individuals who received their first dose of vaccination. Reports also indicate that the older age group had lower neutralizing titers. Infants under the age of 1 are at higher risk of developing severe COVID-19, all infants above the age of 6 months are recommended to take the COVID-19 vaccine. The elderly also have an increased prevalence of adjunct comorbidities which are a significant risk factor to develop severe COVID-19. Various clinical studies have reported that vaccination in these individuals with underlying medical conditions benefits equally to those with no underlying condition, it has been shown that vaccination for COVID-19 has increased benefits compared to its risks [63,64].

## 8. Contraindications of COVID Vaccines

According to the CDC and in agreement with Food and Drug Administration (FDA), European Medicines Agency (EMA), WHO, Medicines and Healthcare products Regulatory Agency (MHRA), immediate and absolute contraindications for all vaccines are as follows [65]:✓Severe anaphylactic reaction following a previous dose or to a component present in the COVID-19 vaccine.✓Immediate allergic reaction of any severity to a previous dose or previously established allergy to a component of the vaccine.

When either of the mRNA vaccines (Pfizer, Moderna) are contraindicated, one might be able to take the (J&J)/Janssen COVID-19 vaccine, or vice versa. Precautionary measures are recommended while taking the COVID-19 vaccine, if there is a previous history of allergic reaction to any other vaccine or previous injectable therapy. Allergies to food, pets, environmental allergies and oral medication allergies are not a contraindication or a precaution to any of the COVID-19 vaccines. In some individuals, delayed onset local reactions have been reported more than a week after the second dose. If the only adverse effect to the vaccine is a delayed onset local reaction, then these individuals can receive subsequent dose of vaccine after the recommended interval, but on the opposite arm [45,50,51,52]. CDC recommends a 15 min observation period for all people after any dose of COVID-19 vaccine, to check for signs of anaphylaxis. In individuals where precautions need to be taken (indications for the same specified above), a 30 min observation period is recommended [65].

## 9. The Booster Dose

### 9.1. The Gap between Two Doses

Several governments altered the time interval between the first and second doses of certain vaccines. The primary rationale was to have the highest number of people receiving the initial dose, with the limited supply of vaccines available.

Studies revealed that Moderna vaccines in a delayed second dose strategy showed a higher success rate in preventing infections when compared to the recommended 28-day interval between doses. However, no such advantage was seen for Pfizer-BioNTech vaccines in reducing infections. Both these were able to reduce hospitalizations and deaths with the delayed second dose. The best outcomes against severe infection were achieved with a dose free interval of 12–15 weeks. A 12-week DSD strategy with a mean efficacy of Moderna vaccines would avert an additional 0.85 (95% CI, 0.62–1.07) hospitalizations and 0.41 (95% CI, 0.33–0.52) deaths/10,000 population, as compared to the recommended vaccination schedule. For Pfizer-BioNTech vaccines, a 12-week delay in second dose administration proved to prevent 0.74 (95% CI, 0.48–1.04) hospitalizations and 0.41 (95% CI, 0.31–0.54) deaths per 10,000 population [66].

Studies on ChAdOx1 nCoV-19 (AZD1222) or COVISHIELD showed that participants aged between 18 and 55 years who waited 12 weeks after the first dose to receive the second, had antibody titers more than 2-fold higher than those individuals who had received the second dose within six weeks of their initial vaccination (geometric mean ratio (GMR) 2.32, 95% CI, 2.01–2.68) [67].

Hence, for mRNA-1273 and BNT162b2 vaccines, increasing dose free interval beyond the recommended 3 and 4 weeks, respectively, could potentially lead to reduced infections, hospitalizations and deaths [54]. For AZD-1222, a dose free interval of 3 months may be preferred over a shorter interval for the roll-out of this vaccine to protect the largest fraction of population in the least amount of time, with a dearth of resources, while also increasing protective efficacy after second dose [67].

An important note on the rapidly mutating Omicron variant, although the currently available vaccines are showing reduced neutralizing capacity to this VOC, its detection in South Africa, where the reach of vaccine to the population is only 7.5%, may show that the vaccines remain protective against it. The variant is more likely to mutate and spread in an area of lower vaccine coverage [68,69]. In multiple preprints published worldwide, it is quickly becoming clear that a two-dose regimen is proving inadequate in its response against the Omicron variant. A three-dose booster regimen has shown a better capacity to protect and reduce morbid disease in individuals diseased with this VOC. Several countries began experiments using different vaccines, such as the J&J, NVX, Moderna and AstraZeneca, after two doses of primary vaccination with mRNA vaccines, showing promising results, especially with the arising VOCs such as the Omicron [46].

### 9.2. Is the Booster a Necessity?

Repeated mutations in the virus and the emerging strains have raised doubts regarding the requirement of administering a booster dose. Currently, the regimens contain two doses of vaccines, offered to be taken with a gap of at least two weeks after the first dose so that there is a development of adaptive immunity; a second dose is required to make sure that the immunity does not fade away and that the immune system can consolidate this protection long term.

Multiple studies over the last year have explored the benefits and detriments of the booster dose in different COVID vaccines as well as the effect of different temporal distances between primary and secondary doses. A randomized, controlled phase-2 trial, the COV-BOOST trial, investigated the reactogenicity and immunogenicity of seven different COVID-19 vaccines as a third dose after two doses of ChAd (Oxford-AstraZeneca) or BNT (Pfizer-BioNtech). It showed greater immunogenicity in the subjects in all seven vaccine studies [70]. In a report studying the Safety and Efficacy of a Third Dose of BNT162b2 COVID-19 Vaccine, a third dose of the vaccine elevated the immune response against COVID-19 by 95.3%, as compared with two doses of the BNT162b2 vaccine [71]. With regard to the AZD1222 booster, Yorsaeng R et al. reported higher immunogenicity in individuals with the booster, as compared to only two doses [72]. Yet, another study demonstrating the effectiveness of a third dose of the inactivated vaccine, it showed that the additional dose augmented the antibody response, increasing the antibodies, where they were falling. Moreover, the neutralizing capacity of these antibodies were far superior to that of antibodies in a two-dose schedule [73].

An Australian study showed that over 250 days, the immunity in individuals started to fade away, and there was a loss of protection against the virus [74]. Recently Pfizer publicly made a reference to data from Israel to conclude that vaccines remained effective against the delta variant as they prevented hospitalization and severe illness. However, German BioNTech announced that there would be a requirement for a booster dose within twelve months of completing the basic COVID regimen to fight against the emerging new strains [75].

A multinational study, the BRACE trial, involving over 6,800 health workers, which explores susceptibility to VOIs, VOCs after COVID-19 vaccines, which could give a clear idea of a requirement of a booster dose. It assesses if the Bacille Calmette-Guérin (BCG) vaccine can help additionally protect against COVID-19. With COVID-19-specific vaccines made available to healthcare workers, BCOS (BRACE COVID-19-Specific vaccine sub-study) will look at if the BCG vaccine would better the immune response to Pfizer, AstraZeneca and CoronaVac vaccines [76]. There are two main views in accordance with this controversy. The first view states that booster doses are required for diseases such as Tetanus, Diphtheria and Pertussis because they help the immune system to remember the previous infection and hence within less time, produce antibodies and work against the infection. The other view states that there is no requirement for a booster dose for the SARS virus, just like for diseases such as Mumps, Rubella and Measles; a booster dose is not given after completing the regimen. The controversy still remains—is the booster dose an absolute necessity?

## 10. What Does the Future Hold?

With the face of this disease evolving every minute, it is crucial that the world of therapeutics continue to grow and discover new and improved vaccines for the everchanging virus. As of 10 November 2022, 64 vaccines are in phase I, 71 in phase II and 91 in phase III of clinical trials. Among the vaccines in phase III are protein subunit vaccines such as Nuvaxovid, in trial in 14 countries and approved in 40, and Razi Cov Pars, in trial in 5 countries and approved in 1. Additionally, in phase III are VLP (Virus-like particle) vaccines such as Covifenz and LYB001, which contain viral proteins which mimic the virus in architecture but have no nuclear material. Some DNA vaccines in phase III are ZyCoV-D and INO-4800, which have a similar mechanism of action as mRNA vaccines. Of note, some companies are developing vaccines against particular VOCs, such as the Omicron COVID-19 Vaccine (Vero Cell) by the China National Biotec Group Company Limited and the Omicront Vero cell vaccine by Sinovac, although clinical efficacy is yet to be documented [77].

## 11. Conclusions

The provision of so many vaccines in such a short amount of time is a huge victory for medical science. Within a year of the emergence of SARS-COV2, the vaccines were prepared, phase III trials instated, and data were available to develop strategies to administer the vaccine.

So far, the efficacy and safety provided by various COVID vaccines is encouraging and can be a way forward to put a hold on this pandemic. The need to ensure current and future pandemic preparedness presents multiple hurdles, among which are equitable vaccine access and the rising trend of vaccine hesitancy at an individual and international level, which are beyond the scope of this discussion.

With all the new vaccines and many still in trials, we are yet to understand the duration of protective immunity the vaccine can provide and how the newly developing variants of the virus will respond to the vaccine. It is well known for viruses that new strains emerge with ongoing transmission, and getting a higher vaccination rate in a population is one of the most important factors to reduce transmission rate and hence to prevent the emergence of new and resistant variants that can escape vaccine immunity. It is premature to say how long the immunity with COVID vaccines will last as trials are still ongoing to see the long-term effectiveness; hence, the booster dose is recommended for high-risk individuals.

The next challenge is to increase the global uptake of vaccines to minimize the transmission. In this era of globalization, no country is protected until we achieve mass immunity in the entire world population. Putting border control measures in place can help to some extent for the short term, but for long-term protection, it is essential that vaccine uptake is high everywhere in the world and not only in developed countries as this will continue to support variants emergence and hence reduce vaccine efficacy. Despite mass production of vaccines, the current stocks may not be able to meet the requirement. Work should be done to enable the low-income countries to become as independent in their vaccine production as possible.

Finally, the amount of resources and investment used for COVID vaccine production and the new vaccine technologies developed should not be reserved for COVID only and should be replicated to combat other infectious diseases.

## Figures and Tables

**Figure 1 diseases-10-00112-f001:**
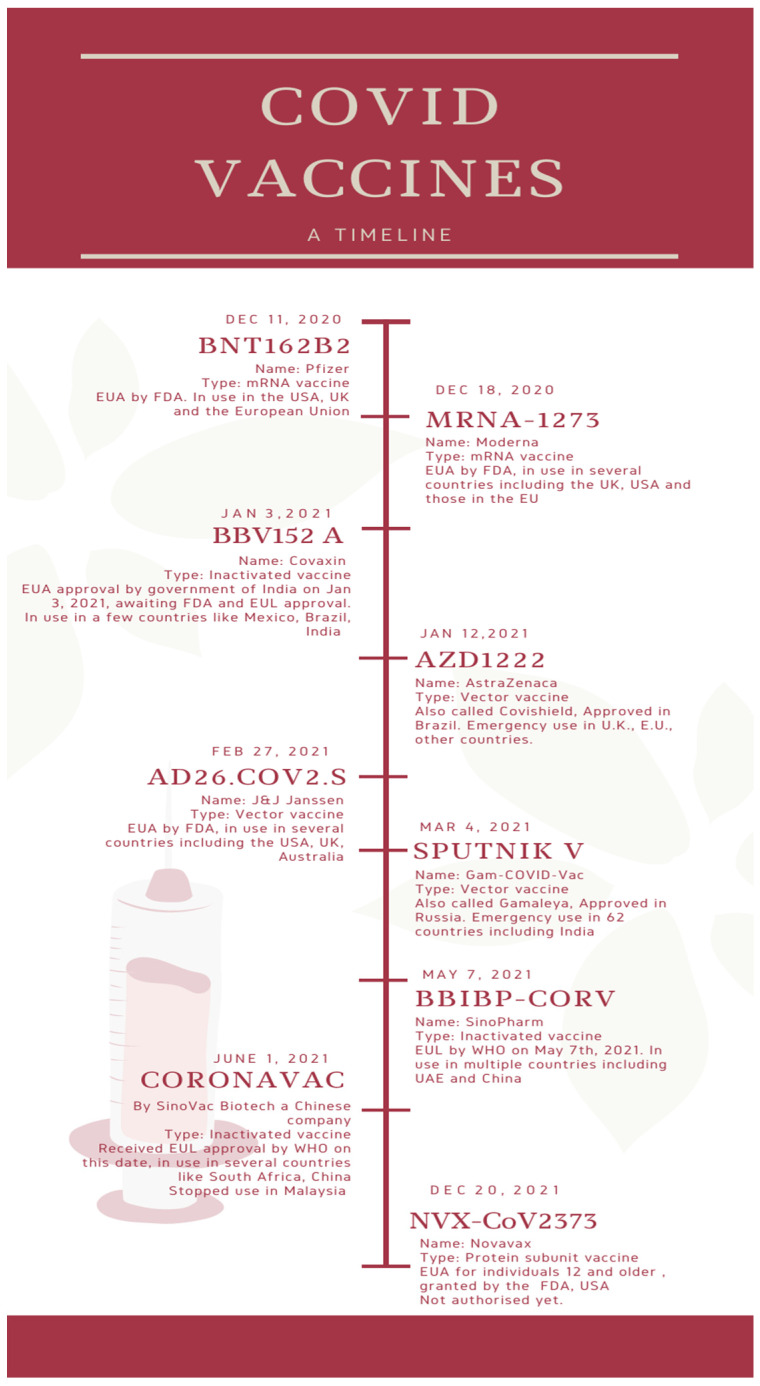
A timeline of the major COVID vaccines [25].

**Figure 2 diseases-10-00112-f002:**
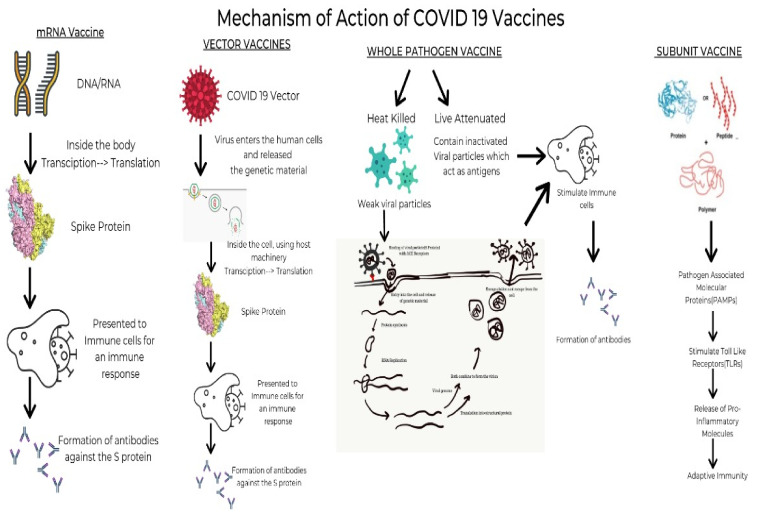
The mechanism of action of various types of vaccines.

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
