# Peer review of "Efficacy and Safety of COVID-19 Vaccines—An Update"

_diseases, 2022, doi:10.3390/diseases10040112_

Round 1

Reviewer 1 Report

The present article presents an overview of the vaccines models developed or in development, in order to establish a vaccination strategy in order to control the Covid-19 pandemic around the world.

Figures are well done and describe the time line (figure 1) of major models of COVID-19 vaccines development. Figure 2 describes the mechanisms of action of various types of vaccines.

References are also in accord with the development of the pandemic and and focus on the diverse vaccines developed or in development.

Author Response

Dear Editor and Reviewers,

I am pleased to resubmit for publication the revised version of the manuscript entitled Efficacy and Safety of COVID-19 Vaccines An update”.

Thankfully the reviewers provided us with a great deal of guidance, regarding how to better position the article. We are hopeful you agree that this revision will update our comprehensive review. All the comments have been addressed, as shown in the revised version of the manuscript, along with this point-by-point response to the reviewers' comments.

All corresponding are blue changes in the manuscript.

REVIEWER 1:

The present article presents an overview of the vaccines models developed or in development, in order to establish a vaccination strategy in order to control the Covid-19 pandemic around the world.

Figures are well done and describe the time line (figure 1) of major models of COVID-19 vaccines development. Figure 2 describes the mechanisms of action of various types of vaccines.

References are also in accord with the development of the pandemic and focus on the diverse vaccines developed or in development.

Reviewer 1: Response

Thank you for your appreciation.

Reviewer 2 Report

This work represents a contribution that is part of a series of other recent reviews and updates publications regarding the most current knowledge on anti-Covid-19 vaccines.

The design of the work is well structured and addresses clearly and effectively on the most important aspects.

The iconographic part offers a useful contribution with the various tables inserted to complete the text.

Two important aspects must be studied in depth: the issue of variants and vaccination protection against them and the issue of use in specific populations.

The first theme is dealt with a clear reference to the different variants that have been gradually registered; actually, a more extensive study could be offered to the theme of the results regarding the different temporal distances in the doses of the boosters and boosters.

 The other aspect is its use in specific populations; as known for some patients, in particular with chronic diseases and / or problems of congenital or acquired immunocompetence.

Author Response

Dear Editor and Reviewers,

I am pleased to resubmit for publication the revised version of the manuscript entitled Efficacy and Safety of COVID-19 Vaccines An update”.

Thankfully the reviewers provided us with a great deal of guidance, regarding how to better position the article. We are hopeful you agree that this revision will update our comprehensive review. All the comments have been addressed, as shown in the revised version of the manuscript, along with this point-by-point response to the reviewers' comments.

All corresponding are blue changes in the manuscript.

REVIEWER 2:

This work represents a contribution that is part of a series of other recent reviews and updates publications regarding the most current knowledge on anti-Covid-19 vaccines.

The design of the work is well structured and addresses clearly and effectively on the most important aspects.

The iconographic part offers a useful contribution with the various tables inserted to complete the text.

Two important aspects must be studied in depth: the issue of variants and vaccination protection against them and the issue of use in specific populations.

The first theme is dealt with a clear reference to the different variants that have been gradually registered; actually, a more extensive study could be offered to the theme of the results regarding the different temporal distances in the doses of the boosters and boosters.

The other aspect is its use in specific populations; as known for some patients, in particular with chronic diseases and / or problems of congenital or acquired immunocompetence.

Reviewer 2: Response

Thank you for the comments, the following changes have been made.

1. Noted. The number of articles referenced in the booster dose section, exploring the themes you have mentioned have been increased and the information has been updated. References 70-73 - “Multiple studies over the last year have explored the benefits and detriments of the booster dose in different COVID vaccines as well as the effect of different temporal distances between primary and secondary doses. A randomized, controlled phase2 trial, the COVBOOST trial, investigated the reactogenicity and immunogenicity of seven different COVID19 vaccines as a third dose after two doses of ChAd (OxfordAstraZeneca) or BNT (PfizerBioNtech). It showed greater immunogenicity in the subjects in all seven vaccine studies [70]. In a report studying the Safety and Efficacy of a Third Dose of BNT162b2 Covid-19 Vaccine, a third dose of the  vaccine elevated the immune response against COVID-19 by 95.3%, as compared with two doses of the BNT162b2 vaccine [71]. With regards to the AZD1222 booster, Yorsaeng R et al reported higher immunogenicity in individuals with the booster, as compared to only two doses [72]. Yet another study demonstrating the effectiveness of a third dose of the inactivated vaccine, it showed that the additional dose augmented the antibody response, increasing the antibodies, where they were falling. Moreover, the neutrilizing capacity of these antibodies were far superior to that of antibodies in a two dose schedule [73].

2. Noted. Conducted a literature review on the same and have added information and references Section 7-COVID Vaccines in Specific Populations” Reference [62]-“The conception of the COVID-19 pandemic was a monumental threat not only to the general population but determinately to a certain sub-section of the population encompassing those with chronic diseases which include autoimmune conditions, those with ongoing immunosuppression or those undergoing cancer treatment. Those patients who are immunocompromised were not recommended to take live attenuated vaccines as the possibility of infection was relatively higher in these patients. Similarly in those patients undergoing cancer treatment, live vaccines were not advised, vaccination against COVID-19 for all cancer patients was recommended and deemed as safe and effective except for those who are currently on anti-B cell therapies, in such patients an interval of four to six months is advised for vaccination post cessation of medication. It is recommended for all those patients who are scheduled for solid organ transplants to be vaccinated before the transplantation; post-transplantation, a minimum of a three months interval should be followed before vaccination to avoid acute rejection. While considering autoimmune conditions, specifically Multiple Sclerosis- vaccination is advocated primarily without cessation of treatment keeping in mind that there could be an imminent risk of worsening or relapse of the disease following discontinuation of therapies. In Irritable Bowel Syndrome (IBS) vaccination is recommended for all patients, live attenuated vaccines are not proffered and for those patients with acute presentations, a gap is recommended before vaccination, when the patient is taking a lower dose of corticosteroids.”.

Reviewer 3 Report

The idea of a brief, concise review with focus on the effectiveness and adverse events of the currently circulating vaccines against COVID-19  highlighting results  split by  different VOCs, is brilliant. However, methodological and linguistic issues prevents from publication under the current form. Please find below my suggestions:

1. The  figure 1 depicting timeline Vaccines, should include also the Sputnik and any other vaccine reported into the text. Tregoning et al ( https://www.nature.com/articles/s41577-021-00592-1) gives a short description of the most  important vaccines that have been approved and already used worldwide.

2. Description of the immune pathways after vaccine administration should be more thoroughly presented. Figure 2 summarised all pathways  in a very comprehensive way, however, relevant text is poor. Moreover, it is important to incorporate data on functional B and T cells following vaccination ( e g doi: 10.1016/j.vaccine.2022.10.017.) as it  might interpretate the short duration of immune protection of  all available vaccines.

3. The bivalent mRNA vaccine ( Nature Medicine Oct 2022) should be mentionned with the caution of preliminary data,  lacking of clinical validation.

4. Data on the efficacy of different vaccines on various VOCs should be carefully addressed. Limitations of studies  (eg data from less than 20 individuals  vs  data from  more than 10,000 ones) should be assessed; not only giving OR (CI95%)  without  reporting the reference population. By adding this information to each subgroup, the reader will get a clearer picture of the burden  of the already published data.

5. Data split by VOCs sould be checked again to ensure the correspondence of references. For example, see lines 224-230.

6. Future approaches including substances of the ongoing phase III trials could be added to the manuscript.

7.Factos associated with negative or positive (or not yet clarified) impact of vaccination could be presented as categories  involving i)the host ( eg immunocompromised, number of comorbidities, aging or neonates, infants, T-cell antigenic response), ii) public health-care issues (eg compliance, number of dose, variability in time intervals of administration, availability, protection policies ,cost), iii) SARS-CoV-2 variants' transmisibility and polymorphisms.     It is useful to summarize relevant data (eg Nature2021;596(7872):417-422.) at the discussion section.

Author Response

Dear Editor and Reviewers,

I am pleased to resubmit for publication the revised version of the manuscript entitled Efficacy and Safety of COVID-19 Vaccines An update”.

Thankfully the reviewers provided us with a great deal of guidance, regarding how to better position the article. We are hopeful you agree that this revision will update our comprehensive review. All the comments have been addressed, as shown in the revised version of the manuscript, along with this point-by-point response to the reviewers' comments.

All corresponding are blue changes in the manuscript.

REVIEWER 3:

The idea of a brief, concise review with focus on the effectiveness and adverse events of the currently circulating vaccines against COVID-19  highlighting results  split by  different VOCs, is brilliant. However, methodological and linguistic issues prevents from publication under the current form. Please find below my suggestions:

1. The figure 1 depicting timeline Vaccines, should include also the Sputnik and any other vaccine reported into the text. Tregoning et al ( https://www.nature.com/articles/s41577-021-00592-1) gives a short description of the most important vaccines that have been approved and already used worldwide.

2. Description of the immune pathways after vaccine administration should be more thoroughly presented. Figure 2 summarised all pathways in a very comprehensive way, however, relevant text is poor. Moreover, it is important to incorporate data on functional B and T cells following vaccination (e g doi: 10.1016/j.vaccine.2022.10.017.) as it might interpret the short duration of immune protection of all available vaccines.

3. The bivalent mRNA vaccine (Nature Medicine Oct 2022) should be mentioned with the caution of preliminary data, lacking of clinical validation.

4. Data on the efficacy of different vaccines on various VOCs should be carefully addressed. Limitations of studies (eg data from less than 20 individuals vs data from more than 10,000 ones) should be assessed; not only giving OR (CI95%) without reporting the reference population. By adding this information to each subgroup, the reader will get a clearer picture of the burden of the already published data.

5. Data split by VOCs should be checked again to ensure the correspondence of references. For example, see lines 224-230.

6. Future approaches including substances of the ongoing phase III trials could be added to the manuscript.

7.Factos associated with negative or positive (or not yet clarified) impact of vaccination could be presented as categories involving i)the host ( eg immunocompromised, number of comorbidities, aging or neonates, infants, T-cell antigenic response), ii) public health-care issues (eg compliance, number of dose, variability in time intervals of administration, availability, protection policies ,cost), iii) SARS-CoV-2 variants' transmisibility and polymorphisms. It is useful to summarize relevant data (eg Nature. 2021;596(7872):417-422.) at the discussion section.

Reviewer 3: Response

Thank you for your appreciation. Your suggestions and comments have been noted. The following changes have been made.

  1. Figure: 1 has been updated and the article linked, has been added in the references list; reference 25

  2. Noted, paper was reviewed, and following information was added. References 17, 18, and 20.

    • [17]-“BNT162b2  developed by Pfizer/BioNtech elicits an immunological response by inducing IgG, IgA, CD8+ cells, or CD4+ cells, while mRNA-1273  developed by Moderna induces CD8 T cell response.”

    • [18]-“Once injected , the immune cells of the body detect presence of foreign antigen and activate an immune response by producing antibody-producing B cells and  T cells that seek out and destroy infected cells. T cells act by examining the storage of proteins expressed on the surfaces of cells. Since they can  recognise the body’s own proteins as ‘self’, if they find a foreign protein, they activate  an immune response against the cell  storing it.”

    • [20]-“Since these vaccines are versions of weak  natural pathogens, the immune system activates a range of defences like killer T cells which identify and destroy infected cells , helper T cells which support antibody production and antibody-producing B cells which will target pathogens.”

  1. Noted. A cautionary statement has been added over the section of efficacy against variants.-“The following section highlights the effectiveness against specific vaccines against different variants of the virus. It is important to mention at this juncture, that the information should be interpreted with caution as due to the urgency of developing vaccines for this morbid disease, several studies even today are lacking adequate sample size, randomization or distribution over ages, and remain quite preliminary.”

  2. Noted. Have added reference population to relevant studies quoted in this section.

  3. References have been checked and verified. Thank you, and apologies.

  4. Section 9-“What does the future hold?” has been added, reference 77-“With the face of this disease evolving every minute, it is crucial that the world of therapeutics continue to grow and discover new and improved vaccines for the everchanging virus. As of November 10, 2022, 64 vaccines are in phase I, 71 in phase II and 91 in phase III of clinical trials. Among the vaccines in Phase III, are protein subunit vaccines like Nuvaxovid, in trial in 14 countries, approved in 40, and Razi Cov Pars, in trial in 5 countries, approved in 1. Also in Phase III are VLP (Virus-like particle) vaccines like Covifenz and LYB001, which contain viral proteins which mimic the virus in architecture, but have no nuclear material. Some DNA vaccines in Phase III are ZyCoV-D and INO-4800, which have a similar mechanism of action as mRNA vaccines. Of note, some companies are developing vaccines against particular VOCs, like the Omicron COVID-19 Vaccine (Vero Cell) by the China National Biotec Group Company Limited and the Omicront Vero cell vaccine by Sinovac, although clinical efficacy is yet to be documented.”

  5. Noted.

    • For host: -“There has been a reported difference in the immune response to the BNT162b2 vaccine post the first dose. In comparison, the elderly (prominently in > 80 years)showed a significantly lower level of antibodies than the younger individuals who received their first dose of vaccination. Reports also indicate that the older age group had lower neutralizing titres. Infants under the age of 1 are at higher risk of developing severe COVID-19, all infants above the age of 6 months are recommended to take the COVID vaccine. The elderly also have an increased prevalence of adjunct comorbidities which are a significant risk factor to develop severe covid-19. Various clinical studies have reported that vaccination in these individuals with underlying medical conditions benefits equally to those with no underlying condition, it has been shown that vaccination for COVID-19 has increased benefits compared to its risks.” References 63-64

    • Also asked to add the public health care concerns, authors felt that this section is beyond the scope of this paper and warrants another literature review, detailing such.

Reviewer 4 Report

This is an important topic and the article contains some useful information but is very fragmented. It is well structured overall but many sections read like a series of facts from a few studies, with little connection or flow between them. Because of the breadth of the topic many sections are very superficial, missing key points and references. Further, many sections cite other reviews rather than original sources. This is reflected by the relatively small number of references (63) for a review of this breadth. 

The English is good overall but there are multiple small grammatical errors throughout that would benefit from editing by a native English speaker.

Specific comments

·      Page 1, line 46 – you mention Asian ethnicity, haematological malignancies and systemic fungal infections as risk factors but there are many others (e.g. diabetes, hypertension, obesity, immune suppression, respiratory diseases). You should mention some of these with references to provide a more balanced background.

·      Page 2, line 61. “due to the absence of S-gene”. This is not correct as the virus contains S-gene. I think you are referring to “S gene dropout”? This is an artefact related to PCR assays and primer mismatch which should be clarified.

·      Page 2, Line 71. “3 of were rather homogenous to VOC B.1.351”. Please correct.

·      Page 2, line 90 “as of 12 September 2022, a total of 12,613,484,608 vaccine doses”. This sentence is incomplete.

·      Page 3, Line 123 “When introduced into the body, these viruses can grow and replicate 123 within the body but cannot infect the individual”. This is not correct. Live attenuated vaccines can infect other people (e.g. polio, varicella) but cause less severe disease. 

·      Page 4 – Figure 1. Please ensure this figure is original. If it has been adapted from elsewhere this needs to be acknowledged.

·      Page 5 – Figure 2. Please ensure this figure is original. If it has been adapted from elsewhere this needs to be acknowledged.

·      Page 5, line 150 “it is certainly a sufficient indicator of their efficacy in the real world.”. This is not necessarily true as vaccines can also induce cell-mediated immunity etc. Antibody titres are useful but not the whole story. Please rephrase.

·      Discussion of adverse effects is very superficial. 

o   The section on Novavax vaccine discusses efficacy, not adverse events!

o   For Pfizer and Moderna there is mention of common adverse events but no mention of myocarditis/pericarditis which is well described

o   For AstraZeneca there is no mention of the mechanism or VITT, which has been well studied.

·      Discussion of the Gap between Two Doses cite very few articles, essentially quoting many statistics in each paragraph then citing a single review article.

·      Page 12, Line 499. “Another Australian study is to be conducted by Brazilian health workers”. This is misleading as it is a multi-national study. The citation is an online post discussing the BRACE study which actually states “Since the trial launched in March 2020, more than 6800 healthcare workers have enrolled across 36 sites in Australia, Brazil, the Netherlands, Spain and the UK”

Author Response

Dear Editor and Reviewers,

I am pleased to resubmit for publication the revised version of the manuscript entitled Efficacy and Safety of COVID-19 Vaccines An update”.

Thankfully the reviewers provided us with a great deal of guidance, regarding how to better position the article. We are hopeful you agree that this revision will update our comprehensive review. All the comments have been addressed, as shown in the revised version of the manuscript, along with this point-by-point response to the reviewers' comments.

All corresponding are blue changes in the manuscript.

REVIEWER 4:

This is an important topic and the article contains some useful information but is very fragmented. It is well structured overall but many sections read like a series of facts from a few studies, with little connection or flow between them. Because of the breadth of the topic many sections are very superficial, missing key points and references. Further, many sections cite other reviews rather than original sources. This is reflected by the relatively small number of references (63) for a review of this breadth. 

The English is good overall but there are multiple small grammatical errors throughout that would benefit from editing by a native English speaker.

Specific comments

  1. Page 1, line 46 you mention Asian ethnicity, haematological malignancies and systemic fungal infections as risk factors but there are many others (e.g. diabetes, hypertension, obesity, immune suppression, respiratory diseases). You should mention some of these with references to provide a more balanced background.

  1. Page 2, line 61. due to the absence of S-gene. This is not correct as the virus contains S-gene. I think you are referring to S gene dropout? This is an artefact related to PCR assays and primer mismatch which should be clarified.

  2. Page 2, Line 71. 3 of were rather homogenous to VOC B.1.351. Please correct.

  3. Page 2, line 90 as of 12 September 2022, a total of 12,613,484,608 vaccine doses. This sentence is incomplete.

  4. Page 3, Line 123 When introduced into the body, these viruses can grow and replicate 123 within the body but cannot infect the individual. This is not correct. Live attenuated vaccines can infect other people (e.g. polio, varicella) but cause less severe disease. 

  5. Page 4 Figure 1. Please ensure this figure is original. If it has been adapted from elsewhere this needs to be acknowledged.

  6. Page 5 Figure 2. Please ensure this figure is original. If it has been adapted from elsewhere this needs to be acknowledged.

  7. Page 5, line 150 it is certainly a sufficient indicator of their efficacy in the real world.. This is not necessarily true as vaccines can also induce cell-mediated immunity etc. Antibody titres are useful but not the whole story. Please rephrase.

  8. Discussion of adverse effects is very superficial. 

    1. The section on Novavax vaccine discusses efficacy, not adverse events!

    2. For Pfizer and Moderna there is mention of common adverse events but no mention of myocarditis/pericarditis which is well described

    3.  For AstraZeneca there is no mention of the mechanism or VITT, which has been well studied.

  1. Discussion of the Gap between Two Doses cite very few articles, essentially quoting many statistics in each paragraph then citing a single review article.

  2. Page 12, Line 499. Another Australian study is to be conducted by Brazilian health workers. This is misleading as it is a multi-national study. The citation is an online post discussing the BRACE study which actually states Since the trial launched in March 2020, more than 6800 healthcare workers have enrolled across 36 sites in Australia, Brazil, the Netherlands, Spain and the UK

Reviewer 4: Response

Thank you for your comments. Following are the changes made:

  1. Risk factors for potentially fatal disease have been added. “ The disease spectrum of this pathogen ranges from a mild self-limiting infection to potentially fatal disease with inflated morbidity and mortality figures for individuals with coexisting morbidities such as diabetes mellitus, obesity and other lifestyle diseases, along with poorer outcomes in patients with respiratory co-morbidities, like COPD.” A reference has been added for the same as well, [5]

  1. Corrected the sentence.” It was shortlisted as a VOC due to the S-gene target failure in PCR samples, as well as the presence of 17 mutations in its genome.”

  2. Corrected.

  3. Corrected. “have been administered”

  4. Corrected. “cannot cause symptomatic disease in the individual.”

  5. Figure:1 used in our paper is completely original, credits: Eshani Sharma.

  6. Figure:2 used in our paper is completely original, credits: Shubham Goyal.

  7. Rephrased. “and certainly doesnt paint the entire picture, barring the effects of T-cell immunity, complement system, it gives a preliminary idea of their efficacy in the real world.”

  8. The adverse effects section has been updated, with the following changes.

    1. This is a gross error on our part; apologies, have updated the section, replaced it with the following: “A study conducted by Health T et al exploring the safety and efficacy of NVX-Co2373, involving about 7500 individuals receiving the vaccine found various adverse effects after both first and second dose, ranging from mild, but more common symptoms like fatigue, headache, muscle pain, to more serious systemic adverse events like a fever (between 38°C to 40°C in 2.4% participants after first dose, and 5.4% after the second dose) of >40°C in 2 participants, one after the first dose, and one after the second. Of note, one serious adverse event, myocarditis, was reported in a participant 3 days after the second dose, identified as a possible immune mediated adverse event. Adverse events were less serious and occurred less frequently in older participants in the trial. Due to the number of participants, it is not possible to exclude the occurrence of the rare adverse events.” Reference 53 has been replaced as well.

    2. Noted. Have added information on the same, with reference number[66].-“Of note, a rare, but serious adverse event was noted in the population of the United States, within its vaccine adverse events reporting system, catching 1226 reports of myocarditis after either dose of these mRNA vaccines, between December 2020-June 2021, with acute hospitalization, but no casualties. This lead to a review by the CDC, but the benefits of the mRNA vaccines remained greater than the risks, and their use continued [55].”.

    3. Have added the occurrence of VITT in Astrazenaca vaccine, have added reference number 64 as well.-“ A more serious adverse event has been described in a case series by Schultz et al, where they reported a rare vaccine related event, vaccine induced thrombotic thrombocytopenia, where 5 health workers within a study population of 130,000 developed serious venous embolism and concurrent thrombocytopenia 7-10 days after the first dose of the vaccine [59].

  1. Noted. Number of articles cited, with additional information have been added. References 70-73 - “Multiple studies over the last year have explored the benefits and detriments of the booster dose in different COVID vaccines as well as the effect of different temporal distances between primary and secondary doses. A randomized, controlled phase2 trial, the COVBOOST trial, investigated the reactogenicity and immunogenicity of seven different COVID19 vaccines as a third dose after two doses of ChAd (OxfordAstraZeneca) or BNT (PfizerBioNtech). It showed greater immunogenicity in the subjects in all seven vaccine studies [70]. In a report studying the Safety and Efficacy of a Third Dose of BNT162b2 Covid-19 Vaccine, a third dose of the  vaccine elevated the immune response against COVID-19 by 95.3%, as compared with two doses of the BNT162b2 vaccine [71]. With regards to the AZD1222 booster, Yorsaeng R et al reported higher immunogenicity in individuals with the booster, as compared to only two doses [72]. Yet another study demonstrating the effectiveness of a third dose of the inactivated vaccine, it showed that the additional dose augmented the antibody response, increasing the antibodies, where they were falling. Moreover, the neutrilizing capacity of these antibodies were far superior to that of antibodies in a two dose schedule [73].

  2. Have corrected the phrase, acknowledged the BRACE trail.”A multinational study, the BRACE trial, involving over 6800 health workers, which explores susceptibility to VOIs, VOCs after COVID-19 vaccines, which could give a clear idea of a requirement of a booster dose. It assesses if the Bacille Calmette-Guérin (BCG) vaccine can help additionally protect against COVID-19. With COVID-19-specific vaccines made available to healthcare workers, BCOS(BRACE COVID-19-Specific vaccine sub-study).”

Round 2

Reviewer 3 Report

No additional comments for the revised version of the submitted manuscript.  Detailed reply to all queries was provided by the authors.

Reviewer 4 Report

The revised manuscript is significantly improved and the authors have addressed all my previous concerns.